# Does Social Media Affect a Patient’s Decision to Undergo Orthognathic Surgery?

**DOI:** 10.3390/ijerph20126103

**Published:** 2023-06-12

**Authors:** Omar Alsuhaym, Ibrahim Aldawas, Fahad Maki, Mohammed Alamro, Khaled Alshehri, Yazeed Alharthi

**Affiliations:** 1Maxillofacial Surgery and Diagnostic Science Department, College of Dentistry, King Saud bin Abdulaziz University for Health Sciences, Riyadh 11426, Saudi Arabia; 2King Abdullah International Medical Research Center, Ministry of National Guard Health Affairs, Riyadh 11481, Saudi Arabia; 3College of Dentistry, King Saud bin Abdulaziz University for Health Sciences, Riyadh 11426, Saudi Arabia

**Keywords:** orthognathic surgery, social media, corrective jaw surgery, surgery decision, maxillofacial surgeon

## Abstract

Orthognathic surgery, also known as corrective jaw surgery, is a procedure that corrects abnormalities of the jaw and face. It is used to treat malocclusions, where the teeth and jaws are misaligned. This surgery can improve the function and appearance of the jaw and face, leading to improved mastication, speech, and quality of life for the patients. To assess if social media had any effect on the patients’ decision to undergo orthognathic surgery, a self-administered online questionnaire was distributed to patients who had undergone orthognathic surgery at the Oral and Maxillofacial department through the health information system (BESTCare, 2.0A) to participate in the study. In total, 111 responses were recorded from the patients, with 107 agreeing to answer the questionnaire and 4 refusing to answer. Twitter was a source of information about orthognathic surgery for 61 patients (57%). When using a social media platform, 3 patients (2.8%) were influenced by an advertisement or an educational post on social media that presented the surgical correction of the jaws, while 15 (14%) believed that they had been somewhat influenced, and 25 (23.4%) picked their surgeon through social media. Fifty-six patients (52.3%) took the neutral position regarding whether information on social media had answered their questions and concerns regarding the surgical procedure. Social media did not influence patients’ decision to undergo the procedure. Surgeons and specialists must utilize their platforms to answer any concerns or questions from any patient undergoing or having undergone this corrective jaw surgery.

## 1. Introduction

Orthognathic surgery is a well-known surgical intervention that entails a series of procedures on the jaws and chin to alter and/or correct facial features [1]. The primary goals of orthognathic surgery are to improve functionality (chewing, mastication, phonetics, and respiration), refine facial aesthetics, and enhance the stability of orthodontic treatment [2]. Patient resources regarding such interventions can create challenges, with the Internet becoming an integral source for patients [3].

Despite the potential benefits of using social media platforms to connect with physicians, the risk of misleading information has become a real concern. The spread of wrong information via social media is a significant issue because anyone can broadcast anything on these platforms without the oversight of specialists or administrators [4]. Therefore, providing psychological influence and sufficient information to the patient is critical during the decision-making process. Adult patients use the Internet to gather information before deciding on orthognathic surgery. Before the procedure, some individuals obtain information from their friends who have had negative experiences with orthognathic surgery, while others obtain information from their maxillofacial surgeons [5,6]. Both systems have advantages and disadvantages, and social media today certainly provides rapid and easy access to knowledge about any topic in medicine or dentistry.

There are currently no reports on the influence of social media on the process of undergoing orthognathic surgery. However, two studies have focused on this matter from a cosmetic surgery perspective. The first study assessed the impact of social media on females visiting King Abdulaziz Medical City in Riyadh, Saudi Arabia [7] as follows: overall, 77% of participants thought that such procedures were affected by the price, 97% felt that specialized accounts on social media were helpful, and 77.8% felt otherwise [7]. They concluded that 68% of participants were influenced by social media, while 31.9% were not [6]. The second study assessed the influence of social media on the decision to undergo cosmetic procedures among female university students [6]. They concluded that 48.5% of participants were influenced by social media to consider undergoing such procedures, while the other participants were not [6]. These types of surgeries can have a positive impact, resulting in a life-changing appearance [6]. However, negative views on the price range can lead the patient to have second thoughts regarding undergoing such procedures [6,7].

The influence of social media grows every day, and millions of individuals rely significantly on their social media accounts for information, to the point where they first seek answers to their medical or dental problems on the Internet rather than speaking with doctors. Therefore, the aim of this study was to examine the impact of social media on people who underwent orthognathic surgery.

## 2. Materials and Methods

This cross-sectional observational study was conducted among people who had attended an oral and maxillofacial department in a dental center. Prior to starting the study, the research protocol was approved by the relevant institutional review board. The data collection process was initiated after obtaining ethical clearance (Ref. No. IRB/1775/22).

### 2.1. Sample Size and Sampling Frame

The sample size required for this study was estimated to have a power of 80% and a 95% confidence interval for a prevalence of 50%, based on previous studies [8,9]. In the authors’ department, the Health Information System (BESTCare, 2.0A) is used to book patients and acquire patients’ information to communicate with them regarding their appointments. This was a significant hurdle in communicating with patients; therefore, the sample size was set to 138, which is the maximum number of patients who underwent corrective jaw surgery.

### 2.2. Inclusion and Exclusion Criteria

Inclusion criteria:Patients who had undergone corrective jaw surgery, either Lefort I osteotomy, sagittal split osteotomy, genioplasty, or a combination of these, between October 2017 and February 2022.Patients who were in the process of undergoing orthognathic surgery (either single jaw or bimaxillary).

Exclusion criteria:Patients who had undergone revision surgery.Patients who had undergone jaw surgery for sleep apnea.

### 2.3. Data Collection

The participants were informed that participation in the study was voluntary and anonymous. The data required for this study were collected using a self-administered, structured questionnaire. The questionnaire was designed after referring to similar studies reported in the literature that were translated into Arabic [9]. The questionnaire comprised a total of 21 questions in three sections; section one consisted of questions related to their demographic details (Q1–Q4), section two consisted of questions related to their use of social media (Q5–Q15), and section three consisted of questions related to the effect of social media on their decision to undergo orthognathic surgery (Q16–Q21).

### 2.4. Validity and Reliability of the Questionnaire

The content validity of the questionnaire was assessed by a panel of experts comprising five oral and maxillofacial surgeons. The main purpose was to determine the questions that had a greater degree of agreement amongst the panel experts and to quantify the concordance between the panel experts. Aiken’s V test was employed for each question [10]. A value higher than 0.80 was obtained for the questions included in this questionnaire.

The reliability of the questionnaire was assessed through conduction of a pilot study among 5% of the population required for this study, in which patients who had undergone surgery and met the inclusion criteria were invited to participate. Test–retest reliability was assessed through the responses obtained from the same participants before and after a gap of 2 weeks. An intraclass correlation coefficient of 8, which indicates good reliability, was obtained. The pilot study responses revealed no need for further modification of the questionnaire; therefore, all 21 questions were included in the final version of the questionnaire. The responses from the pilot study participants were included in the final data analysis.

### 2.5. Participants

Through the Health Information System, patients who had attended the oral and maxillofacial department were invited to participate in the survey. The questionnaire was distributed via messages or by calling the patients. The participants were informed about the study’s aim and that participation in this questionnaire survey was voluntary and anonymous. The data collection process was scheduled over a period of 2 months (from 17 August 2022 to 28 October 2022).

### 2.6. Statistical Analysis

Data were analyzed using SPSS 27.0 (IBM; Armonk, NY, USA). Descriptive statistics were calculated, and a chi-squared analysis was used to compare the effect of social media applications on patients’ decisions to undergo corrective jaw surgery.

## 3. Results

Overall, 111 responses were recorded, with 107 individuals agreeing to participate and 4 who did not. The responses were fewer than the required sample size of 138. The demographic details of the participants are presented in Table 1.

### 3.1. Use of Social Media for Surgery-Related Information

Of the 107 participants who answered the questionnaire (96.4%), Snapchat (Snap Inc.; Santa Monica, CA, USA) was the most used social media application (n = 41, 38.3%), and Telegram (Telegram Messenger Inc.; Tortoga, British Virgin Islands) was the least used (n = 1, 0.9%). The participants mostly spent 2–4 h daily on social media (n = 60, 56.1%), followed by 1–2 h (n = 39, 36.4%), ≥6 h (n = 6, 5.6%), and <1 h (n = 2, 1.9%). Twitter (San Francisco, CA, USA) was considered the most reliable source of information regarding surgical correction of the jaws (n = 61, 57%), followed by Google (Mountain View, CA, USA; n = 18, 16.8%), Instagram (Meta Platforms; Menlo Park, CA, USA; n = 18, 16.8%), WhatsApp (Meta Platforms; n = 7, 6.5%), and Snapchat (n = 1, 0.9%). Two participants (1.8%) provided other answers. Fifty-two (48.6%) participants used a filter to modify their facial appearance while taking a photo, and fifty-five (51.4%) did not. Forty-seven (43.9%) participants were following doctors on social media, and sixty (56.1%) were not. Sixty-five (60.7%) participants had consulted a doctor regarding their jaw condition on social media, whereas forty-two (39.3%) had not. Furthermore, 2 (1.9%) participants strongly agreed that they constantly found updated information regarding surgical correction of the jaws on social media, 29 (27.1%) agreed with this statement, 30 (28%) were neutral, 45 (42.1%) disagreed, and 1 (0.9%) strongly disagreed. Overall, 3 (2.8%) participants had been influenced by someone famous, an influencer, or an advertisement on social media that featured surgical correction of the jaws, 15 (14%) were somewhat influenced, 23 (21.5%) were somewhat not influenced, and 66 (61.7%) had never been influenced. Additionally, 1 (0.9%) participant strongly agreed that they believed everything on social media about how safe and secure the surgery was, 46 (43%) agreed, 31 (29%) were neutral, 26 (24.3%) disagreed, and 3 (2.8%) strongly disagreed. Three (2.8%) participants strongly agreed that social media applications promote surgical correction of the jaws. Finally, 34 (31.8%) participants strongly agreed that social media is a strong tool to influence someone’s decision to undergo surgical correction of the jaws, 44 (41.1%) agreed, 16 (15%) were neutral, 11 (10.3%) disagreed, and 2 (1.9%) strongly disagreed (Table 2).

### 3.2. Social Media and Patients Who Had Undergone Surgical Correction of the Jaws

Of all 107 participants who answered the questionnaire, 25 (23.4%) selected their surgeon through social media and 82 (76.6%) did not. Moreover, for the patients that picked their surgeon through social media, 16 (64%) chose their surgeon for their expertise in surgical correction of the jaws, 5 (20%) chose their surgeon because of their activity on social media, and 4 (16%) provided other reasons. Furthermore, 9 (8.4%) participants agreed that social media had helped improve their quality of life, 16 (15%) were neutral, 76 (71%) disagreed, and 6 (5.6%) strongly disagreed. Regarding finding information on social media that answered their questions and concerns, 4 (3.7%) participants strongly agreed that they had found helpful information, 28 (26.2%) agreed, 56 (52.3%) were neutral, 16 (15%) disagreed, and 3 (2.8%) strongly disagreed. 

There were 15 (14%) participants who agreed that they found postoperative instructions on social media, 15 (14%) were neutral, 58 (54.2%) disagreed, and 19 (17.8%) strongly disagreed. Finally, 3 (2.8%) participants strongly agreed that they found information on postoperative complications via social media, 24 (22.4%) agreed, 19 (17.8%) were neutral, 40 (37.4%) disagreed, and 21 (19.6%) strongly disagreed (Table 3). 

### 3.3. Impact of Social Media on Patient’s Decision to Undergo Orthognathic Surgery

Most female respondents (52, 67.5%) consulted a doctor through social media about their jaw condition (χ^2^ = 5.3; *p* = 0.021), and most females used a face filter when taking pictures to modify their appearance (χ^2^ = 39.41; *p* = 0.001). However, 45 (42.1%) of the overall participants disagreed with the availability of updated information about orthognathic surgery on social media platforms. Most male respondents (25, 83.3%) who had surgical correction of their jaws had never been influenced by someone famous, an influencer, or an advertisement on social media (χ^2^ = 9.16; *p* = 0.027), although 46 (43%) respondents believed any information present on social media related to the safety of the procedure. Overall, 80 (74.8%) participants thought that orthognathic surgery was not well-promoted on social media, but 78 (70.2%) participants believed in the power of social media and its effect on the decision to undergo surgery. The participants also reported that social media was not beneficial in terms of information on postoperative instructions and complications. Additionally, there were no significant differences in answers by sex, marital status, educational level, or age; thus, a correlation between the strength of social media and its effect on patients’ decisions has not been established (Table 4).

## 4. Discussion

Social media platforms have become a major part of our lives in the 21st century. They have changed the way we communicate, shop, and even how we view ourselves. With the rise of social media, it has become easier to compare ourselves with others and strive for “perfection.” This can lead to an increase in the number of people wanting to undergo orthognathic surgery, a corrective jaw surgery that can improve facial appearance. In the present study, the impact of social media on people’s decisions to undergo orthognathic surgery and its potential risks and benefits were further explored.

The use of social media has been linked to increased levels of body dissatisfaction among adolescents and young adults [11]. This dissatisfaction is often caused by comparing oneself to others on social media platforms. The most commonly used platform is Instagram [12]. Some individuals may feel pressured to look a certain way or have certain facial features that are seen as desirable on these platforms. This can lead them to change their desired appearance [13].

However, potential risks associated with orthognathic surgery should be considered before the procedure. These symptoms include pain, swelling, infection, nerve damage, and scarring [14]. Additionally, there is a risk of not achieving the desired outcome or having an asymmetrical result [14]. Therefore, it is important for individuals to consider orthognathic surgery as one of several options and to be aware of these potential risks before considering the procedure. 

There are potential benefits associated with orthognathic surgery. These include improved facial symmetry, aesthetics, and function, such as improved speech or chewing ability [15]. Additionally, some studies have found that patients who undergo orthognathic surgery experience improved self-esteem after the procedure [16]. Therefore, it is important for individuals to consider orthognathic surgery and weigh both the risks and benefits before making such a major decision.

### 4.1. Social Media and Orthognathic Surgery

Coleman et al. analyzed the use of social media among orthognathic surgery patients and found that patients in their early twenties searched social media to obtain information regarding orthognathic surgery [17]. In contrast, in the present study, patients in their late twenties mostly searched social media to obtain information regarding orthognathic surgery. In terms of looking for information on orthognathic surgery, Buyuk and Imamoglu reported that most posts uploaded to Instagram were about patients’ experiences (49.1%), making social media an inadequate source of information regarding orthognathic surgery [13]. However, in the present study, most participants found that posts on Twitter (n = 61, 57%) were most useful in acquiring information about the surgery. This may be because many posts on Twitter are by specialists. Similarly, Bhamrah et al. reported the use of Internet information by patients undergoing orthognathic surgery and found that most patients were utilizing social media to look for posts about previous experiences, which shows the need for surgeons on social media to educate patients about the procedure and answer patients’ questions [18]. This suggests that the role of surgeons on social media should not be underestimated. In addition, Findik and Buyukcavus evaluated content posted on Instagram and YouTube about surgery as a first approach for patients undergoing orthognathic surgery and concluded that both platforms provided an insufficient source of information for patients and that most of the content posted was based on previous patients’ experiences and not educational content from specialists [19]. This indicates that surgeons should pay more attention to the information they post on social media platforms, as they serve as a primary source of information for patients seeking orthognathic surgery.

### 4.2. Social Media as a Source of Information after Surgery

Postoperative instructions and information are important in achieving good treatment outcomes and minimizing postoperative complications, and most websites and social media platforms that address postoperative instructions regarding pain control and management are considered poor because they come from unreliable websites that are not composed by surgeons or specialists [20,21]. Similarly, the participants in the present study were dissatisfied with information on social media regarding postoperative complications and instructions. Most information on social media about orthognathic surgery is considered insufficient in terms of quality, and patients cannot rely on it [21].

### 4.3. Impact of Social Media on Orthognathic Surgery Decisions

Social media has had a profound impact on people’s decisions to undergo orthognathic surgery [17]. The prevalence of filtered photos and airbrushed images on social media platforms can lead people to feel inadequate or insecure about their own appearance, leading them to seek ways to improve their appearance [22]. This is especially true for adolescents, who are more likely to compare themselves with the unrealistic standards of beauty presented on social media [23]. Additionally, many influencers have undergone this type of surgery and have shared their experiences online, which may encourage others to do the same [18,24]. This can lead to an increase in the demand for orthognathic surgery, as people strive for a “perfect” look. Furthermore, the accessibility of information on orthognathic surgery on social media has made it easier for potential patients to research and learn more about the procedure [18,24]. This increased knowledge may raise their likelihood of pursuing the procedure.

In the present study, social media did not influence the participants’ decisions to undergo the surgery, which may be attributed to some limitations. Of note, all participants in the present study were from one governmental hospital in an area where the patients had a limited choice of surgeons, which may explain why an association between social media and orthognathic surgery was not established. Another limitation was the lack of existing studies on the relationship between orthognathic surgery and social media with which to compare our results.

As a recommendation for future research, it is better to identify patients who have undergone surgery in the private sector. Additionally, it would be beneficial to include a larger sample size to increase the power of the study. Furthermore, it would be interesting to investigate how different social media platforms (e.g., Facebook, Twitter, and Instagram) influence patient decisions regarding orthognathic surgery. It would also be beneficial to explore how different demographic factors (e.g., age and sex) influence patient decisions regarding orthognathic surgery. Finally, future studies should consider other factors that may influence patient decisions, such as the cost and availability of services.

## 5. Conclusions

Social media is rapidly evolving into one of the most important direct-to-consumer marketing routes [20]. Patients often use social media to find surgeons and interact with them regarding procedures, results, and their experiences in the field. A surgeon’s social media presence can significantly boost their reputation as an expert and demonstrate their style and approach to patients. In the present study, social media did not have an impact on the participants’ opinions regarding orthognathic surgery, and the participants felt that valuable information about the postoperative journey was not found on social media aft er the procedure. To conclude, oral and maxillofacial surgeons have a responsibility to not only provide medical information about a surgical procedure, but also to educate the public on the complete process, including potential risks and complications. They should be available to answer questions and provide a complete and accurate picture to their patients. Effective communication with the public requires a proactive and responsive approach.

## Figures and Tables

**Table 1 ijerph-20-06103-t001:** Demographic details of the participants.

Parameter	Sub-Parameter	Number	Percentage
Age	21–25 years	25	23.4%
	26–30 years	36	33.6%
	30 years and above	46	43%
Sex	Male	30	28%
	Female	77	77%
Education	Bachelor’s degree	73	68.2%
	High school diploma	34	31.8%
Marital status	Married	46	43%
	Single	61	57%

**Table 2 ijerph-20-06103-t002:** Use of social media for surgery-related information.

Questions	Answers		
What social media platform do you use the most?	Instagram	Twitter	Snapchat
	5 (4.7%)	14 (13.1%)	41 (38.3%)
	Telegram	TikTok	WhatsApp
	1 (0.9%)	36 (33.6%)	10 (9.3%)
How much time do you spend on social media?	<1 h	1–2 h	2–4 h
	2 (1.9%)	39 (36.4%)	60 (56.1%)
	≥6 h		
	6 (5.6%)		
What is the social media platform that you see as a source of information regarding surgical correction of the jaws?	Instagram	Google	Snapchat
	18 (16.8%)	18 (16.8%)	1 (0.9%)
	Twitter	WhatsApp	Others
	61 (57%)	7 (6.5%)	2 (1.8%)
Do you use filters when taking a picture to modify your facial appearance?	Yes	No	
	52 (48.6%)	55 (51.4%)	
Do you follow doctors on social media platforms?	Yes	No	
	47 (43.9%)	60 (56.1%)	
Have you ever consulted a doctor through social media about a condition that you had in your jaws?	Yes	No	
	65 (60.7%)	42 (39.3%)	
I constantly find updated information regarding the surgical correction of jaws on social media.	Strongly agree 2 (1.9%)	Agree 29 (27.1%)	Neutral 30 (28%)
	Disagree 45 (42.1%)	Strongly disagree 1 (0.9%)	
I have been influenced by someone famous, an influencer, or by an advertisement on social media that had surgical correction of the jaws.	I have been 3 (2.8%)	Somewhat been 15 (14%)	
	Somewhat not 23 (21.5%)	I have never been 66 (61.7%)	
I believe everything on social media about the safety and security of surgeries that involve correction of the jaws.	Strongly agree 1 (0.9%)	Agree 46 (43%)	Neutral 31 (29%)
	Disagree 26 (24.3%)	Strongly disagree 3 (2.8%)	
Social media promotes surgical correction of the jaw.	Strongly agree 3 (2.8%)	Agree 21 (19.6%)	Neutral 39 (36.4%)
	Disagree 41 (38.3%)	Strongly disagree 3 (2.8%)	
Social media is a strong tool to influence someone’s decision to undergo surgical correction of the jaw.	Strongly agree 34 (31.8%)	Agree 44 (41.3%)	Neutral 16 (15%)
	Disagree 11 (10.3%)	Strongly disagree 2 (1.9%)	

**Table 3 ijerph-20-06103-t003:** Social media and patients who had undergone surgical correction of the jaws.

Questions	Answers			
Did you pick your surgeon through social media?	Yes		No	
	25 (23.4%)		82 (76.6%)	
If yes, why did you choose the surgeon?	His/her expertise in surgical correction of the jaws		His/her activity on social media	
	16 (64%)		5 (20%)	
	Others			
	4 (16%)			
Social media helped me improve my quality of life through surgical correction of the jaws.	Agree		Neutral	
	9 (8.4%)		16 (15%)	
	Disagree		Strongly disagree	
	76 (71%)		6 (5.6%)	
I have found information on social media regarding my concerns and questions about surgical correction of the jaws.	Strongly agree	Agree		Neutral
	4 (3.7%)	28 (26.6%)		56 (52.3%)
	Disagree		Strongly disagree	
	16 (15%)		3 (2.8%)	
I have found information on social media regarding postoperative instructions after surgical correction of the jaws.	Agree		Neutral	
	15 (14%)		15 (14%)	
	Disagree		Strongly disagree	
	58 (54.2%)		19 (17.8%)	
I have found information on social media regarding postoperative complications after surgical correction of the jaws.	Strongly agree	Agree		Neutral
	3 (2.8%)	24 (22.4%)		19 (17.8%)
	Disagree		Strongly disagree	
	40 (37.4%)		21 (19.6%)	

**Table 4 ijerph-20-06103-t004:** Association between the impact of social media and patients’ decision to undergo the surgery.

		What Is the Social Media Platform That You Use the Most?
How much time do you spend on social media?	Chi-square	151.295
	df	24
	Sig.	0.000 *^,a,b^
What is the social media platform that you see as a source of information regarding surgical correction of the jaws?	Chi-square	41.491
	df	42
	Sig.	0.493 ^a,b^
Do you use a face filter when taking a picture to modify your facial appearance?	Chi-square	143.282
	df	12
	Sig.	0.000 *^,a,b^
Do you follow doctors on social media platforms?	Chi-square	119.357
	df	12
	Sig.	0.000 *^,a,b^
Have you ever consulted a doctor through social media about a condition that you had in your jaws?	Chi-square	115.176
	df	12
	Sig.	0.000 *^,a,b^
I constantly find updated information regarding surgical correction of the jaws on social media.	Chi-square	147.117
	df	30
	Sig.	0.000 *^,a,b^
I have been influenced by someone famous, an influencer, or an advertisement on social media that featured surgical correction of the jaws.	Chi-square	125.755
	df	24
	Sig.	<0.001 *^,a,b^
I believe everything on social media about how safe and secure the surgeries that involve correction of the jaws are.	Chi-square	130.616
	df	30
	Sig.	<0.001 *^,a,b^
Social media promotes surgical correction of the jaws.	Chi-square	126.007
	df	30
	Sig.	<0.001 *^,a,b^
Social media is a strong tool to influence someone’s decision to undergo surgical correction of the jaws.	Chi-square	149.142
	df	30
	Sig.	0.000 *^,a,b^
Did you pick your surgeon through social media?	Chi-square	118.159
	df	12
	Sig.	0.000 *^,a,b^
If yes, what did you choose the surgeon for?	Chi-square	33.748
	df	30
	Sig.	0.291 ^a,b^
What is the social media platform that you use the most as a source of information regarding surgical correction of the jaws?	Chi-square	250.689
	df	54
	Sig.	0.000 *^,a,b^
Social media helped me improve my quality of life through surgical correction of the jaws.	Chi-square	127.514
	df	24
	Sig.	<0.001 *^,a,b^
I have found information on social media regarding my concerns and questions about surgical correction of the jaws.	Chi-square	131.340
	df	30
	Sig.	<0.001 *^,a,b^
I have found information on social media regarding post-operative instructions after surgical correction of the jaws.	Chi-square	123.745
	df	24
	Sig.	<0.001 *^,a,b^
I have found information on social media regarding post-operative complications after surgical correction of the jaws.	Chi-square	158.833
	df	30
	Sig.	0.000 *^,a,b^

Results are based on nonempty rows and columns in each innermost sub-table. *. The chi-square statistic is significant at the 0.05 level. ^a^ More than 20% of cells in this sub-table have expected cell counts of less than five. Chi-square results may be invalid. ^b^ The minimum expected cell count in this sub-table is less than one. Chi-square results may be invalid.

## Data Availability

Not applicable.

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
