# Peer review of "Does Social Media Affect a Patient’s Decision to Undergo Orthognathic Surgery?"

_ijerph, 2023, doi:10.3390/ijerph20126103_

Round 1

Reviewer 1 Report

Answer the questions in the attached file, please.

Author Response

1: we agree with the comment, and we have adjusted the sentence to make it not more about
how to make the medical professionals post more advertisements, but to answer any concerns
within their limits online.
2: we deleted the part where we say that “it shorten the duration of orthodontic treatment”,
becuase it scientificaly not proven and there is not a supporting bibliography for it.
3: we agree with your input, there is no benefits at all from searching health information online,
we deleted the part and adjusted the senetnce and adding supporting bibliography
4: we added two supporting bibliography, where they reported that some of their participants
had that ocuurence
5: we specified the inclusion criteria from a surgcial prespective, on which type of surgeries that
patients had. we can specifiy it and say “dentofacial deformity”, but that would have a broad
meaning that would involve patients with tumors or any inflammatory disaeases and can also
include patients who had surgeries for sleep apnea which is in our exclusion criteria, so our focus
was cosmetically based and also with the combined treatment with orthodontics
6: bibliography number 9 is cited, (Abdullah WA. Changes in quality of life after orthognathic
surgery in Saudi patients. Saudi Dent J. 2015;27(3):161–164. doi: 10.1016/j.sdentj.2014.12.001.
Epub 2015 Apr 25.)
7-8-9-10: we apologize fo the unorginiazed manner in table 2, we adjusted them and orginized it
in more orderly manner.
11: We collected the gender for demographic purpose only. A subdivision analysis based on
gender didn't show significant result so it was omitted to minimize the distraction from the main
results of the study.
12: layout corrected in table 4.
13: we adjusted the conclusion and made it more precise and organized in terms of the idea

we truly appreciate your valuable review 

Reviewer 2 Report

The tables 2, 3 and four should be reorganized and more compact so it will be easier to follow the data

Does the questionnaire was evaluated by a statistician? 

Does the questionnaires that were used to construct the one utilized to collect data for this paper follow the same purpose? Did they had the same output and results? 

Author Response

the tables are adjusted in the attached file

Yes, the questionnaire was done and evaluated by a professional biostatistician, and also the questionnaire that was constructed followed the same purpose and had the same output and results 

thank you for your valuable review and comments 

Reviewer 3 Report

interesting work from a common aspect, my question is that how the questionnaire represents the real usage of social media? as we known that the quality and appearance of social advertising can largely affect on the patient choice. 

Author Response

we appreciate your comment and input, we focused from the questionnaire that it represents the usage of social media today from an overall perspective ( usual usage like going through apps, trends, ...) and from how the patient wants to see it from and a medical perspective (for example, more interaction with physicians) 

Round 2

Reviewer 1 Report

thank you for your corrections